# On the Influence of Heat Input on Ni-WC GMAW Hardfaced Coating Properties

**DOI:** 10.3390/ma16113960

**Published:** 2023-05-25

**Authors:** Jan Pawlik, Michał Bembenek, Tomasz Góral, Jacek Cieślik, Janusz Krawczyk, Aneta Łukaszek-Sołek, Tomasz Śleboda, Łukasz Frocisz

**Affiliations:** 1Faculty of Mechanical Engineering and Robotics, AGH University of Science and Technology, A. Mickiewicza 30, 30-059 Krakow, Poland; tgoral@agh.edu.pl (T.G.); cieslik@agh.edu.pl (J.C.); 2Faculty of Metals Engineering and Industrial Computer Science, AGH University of Science and Technology, A. Mickiewicza 30, 30-059 Krakow, Poland; jkrawcz@agh.edu.pl (J.K.); alukasze@metal.agh.edu.pl (A.Ł.-S.); sleboda@agh.edu.pl (T.Ś.); lfrocisz@agh.edu.pl (Ł.F.)

**Keywords:** hardfacing, GMAW, FCAW, C45 steel, heat input, macrocrystalline tungsten carbides, nickel matrix

## Abstract

Hardfacing is one of the techniques used for part lifecycle elongation. Despite being used for over 100 years, there still is much to discover, as modern metallurgy provides more and more sophisticated alloys, which then have to be studied to find the best technological parameters in order to fully utilize complex material properties. One of the most efficient and versatile hardfacing approaches is Gas Metal Arc Welding technology (GMAW) and its cored-wire equivalent, known as FCAW (Flux-Cored/Cored Arc Welding). In this paper, the authors study the influence of heat input on the geometrical properties and hardness of stringer weld beads fabricated from cored wire consisting of macrocrystalline tungsten carbides in a nickel matrix. The aim is to establish a set of parameters which allow to manufacture wear-resistant overlays with high deposition rates, preserving all possible benefits of this heterogenic material. This study shows, that for a given diameter of the Ni-WC wire, there exists an upper limit of heat input beyond which the tungsten carbide crystals may exhibit undesired segregation at the root.

## 1. Introduction

Wear and tear of mechanisms and machine parts is a phenomenon constantly accompanying the operation of any machinery and it is causing measurable financial losses [1]. Deterioration processes usually begin in the superficial layer of a given part [2]. The consequence of wear is a progressive loss of mass and initial, nominal dimensions [3,4]. The technical quality of the part decreases gradually and at some point, the wear processes can exclude the proper operation. Although wear can be caused by abrasion, erosion, adhesion, surface fatigue, and tribochemical reactions, or a combination of these different processes, the major share of wear processes is caused by abrasion [5,6].

Reconstruction of worn parts can be carried out by welding methods, especially surfacing methods, often referred to as hardfacing. Depending on the thickness of the layer to be applied, different processes are used, e.g., gas-shielded consumable electrode surfacing [7,8,9], tungsten inert gas coating [10], plasma surfacing [11], or even electron beam overlaying [12] using suitable filler materials.

Among the available techniques, Gas Metal Arc Welding (GMAW) and Flux-Cored-Arc-Welding (FCAW) seem to be most suitable for most applications, since on the one hand, they offer one of the highest deposition rates (at 2–30 kg of material per hour) [13,14,15] and on the other, equipment and filler material provide cost-effectiveness [16]. Nowadays, new types of cored wires are being utilized in FCAW technology [17,18]. These materials are able to provide a wear-resistant surface that has a much higher abrasion resistance than the substrate material, for instance, structural steel [19,20] (Figure 1). In most cases, in contrast to welding, during hardfacing, only a small dilution of the surfacing material with the base material is required in order to retain the specific properties of the surfacing material. The correct method of choice and process parameters is a key issue to achieve the lowest possible mixing of the surfacing material with the substrate material. The results of a study in publications [21,22,23,24] consider the influence of arc current, voltage, welding speed, wire feed rate, as well as the angle of the welding gun relative to the workpiece being surfaced, on the quality of the surfacing. They propose lower currents and an appropriate welding speed to reduce the mixing of the surfacing with the substrate material. However, the correct choice of these parameters must be proposed individually for each application and material.

In case of GMAW/FCAW hardfacing, the most important parameters are the following:welding speed, being a key factor to the overall heat input coefficient,voltage, being responsible mainly for weld bead geometry and root depth,wire feed, being a main constituent of the current,free stick-out of the welding wire, being a secondary factor for the bead geometry,shielding gas type and flow.

There was a multitude of attempts to make a general model for geometry prediction [25,26,27], yet this time there exists no universal recipe for predicting the weld bead geometry and other properties (such as hardness), as every filler material can behave differently, especially if applied on dissimilar base materials. Hence, the authors claim that every material needs its own series of experiments to build a proper model.

While most of the filler materials have their resultant hardness dependant on the metallurgical phenomena occurring during the material application and cooling [28,29], there exist some types of nickel-based cored wires, that rely their hardness mostly on the solid-phase crystals, for instance, macrocrystalline tungsten carbides. Those crystals are located within the nickel sheath of the welding wire and their distribution and segregation in the fabricated weld bead is influenced by the input hardfacing parameters [30]. The authors of the aforementioned paper achieved plausible results, however, they did not attempt to maximize the process efficiency, as their wire feed was set to approx. 1.75 m/min. The intention of this study was to examine material behavior applied with higher deposition rates. The deposition rate is a parameter dependent on the combination of filler wire diameter, wire feed, and welding speed, and furthermore it is inversely proportional to the heat input parameter.

In the current study, the C45 steel samples were coated with singular stringer weld beads, applied with variable heat input and voltage parameters. This steel is widely used in applications requiring higher strength and hardness than basic construction steel, such as shafts, screws, knives, and other tools, including mining consumables (Figure 1). The samples were cut into slices, etched and measured via a digital microscope. Additionally, in order to estimate the utility features and tungsten carbide segregation, HV05 Vickers hardness was measured in 37 points of every cross-sectional cutout of each sample. Other hardness scales would offer only averaged hardness values, consisting of nickel matrix and WC crystals.

## 2. Materials and Methods

### 2.1. Setup and Specimen

As mentioned in the introductory section, the current study examines the behavior of the nickel-based tungsten carbide cored filler wire applied on C45 (DIN 1.0503/ISO 683-1:1987) steel bars with three different voltage settings and three deposition rate levels. By “deposition rate”, the authors mean the volumetric feed of the filler material, being a product of filler material diameter, wire feed, and hardfacing speed. The studied cored wire consists of a thin-walled seamed nickel tube with partially loose admixture of pre-manufactured tungsted carbide crystals.

The steel material was cut and further grinded with cooling to 125.0 × 25.0 × 15.0 mm in order to ensure no disruption from superficial contamination from oxides, etc. The stringer weld beads of 90.0 mm in length were applied onto room temperature base material and cooled in free-air (Figure 2). In order to measure the dimensional properties and cross-sectional hardness of the hardfaced layers, the samples were cut and polished, as shown in Figure 3. The cutting was performed on Struers Labo-Tom 5 metallographic cutter (Struers, Copenhagen, Denmark) and polished on Struers Labo-Pol 4 (Struers, Copenhagen, Denmark) with 220–800 grit discs and 6–1 µm diamond suspension.

The setup for the hardfaced layer application consisted of Kemppi X8 power source (Kemppi, Lahti, Finland) conjoined electronically with a numerically controlled cartesian hardfacing machine, designed and built by Jan Pawlik (the author) and located at AGH University of Science and Technology (Figure 4). This machine offers a 320 × 300 × 170 mm work area and allows the operator to precisely control the welding speed (0.1–2000 mm/min) and wire feed (1–20 m/min) and is controlled via g-code commands by Duet2Wifi motherboard (Duet, Peterborough, UK) with modified software.

### 2.2. Base and Filler Material Chemical Composition

The material selected for being the base metal was the C45 medium carbon steel, being also an analogue of AISI 1045 or 1.0503 steel. The chemical composition of that steel, provided by the supplier, is presented in Table 1 [31]. The carbon equivalent value for this type of material lays between 0.52–0.82 (depending on the chosen C_e_ calculation standard), which means that the weldability is poor. This creates a possibility for martensitic structures to occur in the Heat Affected Zone (HAZ), which will be inspected in the current study. 

The filler metal chosen for this experiment was a Φ 1.6 mm metal cored, gas-shielded hardfacing wire named Endotec DO*611x, manufactured by Castolin Eutectic (Menomonee Falls, WI, USA). A cross sectional image can be seen in Figure 5. This material is intended to be used in applications, where the parts are subjected to heavy loads and severe abrasion, such as conical picks [32], excavator buckets [33], augers and drills [34,35], etc. The chemical composition of the cored wire, supplied by the manufacturer, is available in Table 2.

The study of chemical composition of the molten wire was carried out on a Foundry-Master (WAS) optical emission spectrometer (Hitachi, Tokyo, Japan). The measurement points were selected randomly in the weld bead cross-sectional area and averaged.

### 2.3. Hardfacing Process Parameter Assumptions

The cross-sectional geometry of the weld bead is dependent on many separate factors, however, the general value of the transectional area is determined mainly by the three factors: wire diameter, wire feed, and welding speed. Since the wire diameter is invariant throughout the process, the experiment relied on adjusting the two remaining parameters. 

The authors wanted to fabricate a relatively thin coating with dimensions as close as possible to 2 mm ± 20% in height and 10 mm ± 20% in width. Taking into consideration the rectangular affinity of an ellipse and assuming that the stringer weld bead resembles a half-ellipse cut-out, one can implement the following Formula (1) [28]:(1)$$\frac{D_{w}{}^{2}}{h\times w}=\frac{v_{b}}{v_{w}}$$
where:
*D_w_*—diameter of the wire [mm] (here: 1.6 mm),*h*—assumed height of the bead [mm] (here: 2 mm),*w*—assumed width of the bead [mm] (here: 10 mm),*v_b_*—velocity of welding head [m/s],*v_w_*—wire feed [mm/min].


This Equation (1) allows to roughly assume the output geometry of the weld bead, however, it enables the operator only to pick a certain velocity ratio; those velocities have to be later adjusted to the process and machine limits (e.g., hardly any power source is capable of supplying enough energy to melt the metal wire conveyed with speed exceeding 30 m/min). Three distinctive levels of the welding speed/material feed ratio were assessed and collected in Table 3.

An additional process parameter, which is responsible mainly for the weld bead width (but indirectly also other dimensions), is the voltage. The voltage levels were based within the range recommended by the manufacturer [36]. All the experimental parameters, along with the experiment design, are randomized and aggregated in Table 3.

Other technological parameters were kept constant, regardless of the experimental run. The wire diameter was Φ 1.6 mm, the free stick-out was 15 mm. The shielding gas was 99% pure argon with flow at approx. 16 l/min. This inert gas was used in order not to introduce any effects of oxidation to the study.

### 2.4. Geometric Measurements of the Weld Bead Dimensions 

The measurements of the applied coatings geometry were performed in the following manner: each steel bar was cut into four pieces, where the first cut was located 35 mm from the one end of the bar, whilst the last was cut 44 mm from the other end. Every set of parameters was represented by each side of two 20 mm cut-outs from the middle.

The prepared and polished cross-sectional cut-outs have been subjected to a series of geometrical measurements: weld bead width (*w*), height (*h*), depth of penetration (*d*), and area above (*A_w_*) and below (*A_p_*) the line designated by the base material top surface. The measurements were carried out with the aid of a Keyence VHX-7000 digital microscope (Keyence, Osaka, Japan) and an exemplary measurement is presented in Figure 6.

### 2.5. Hardness Measurements

The material hardness was measured with a Vickers scale on a Stuers Dura-Scan 4 (Struers, Copenhagen, Denmark) microhardness measurement device. The HV5 hardness was measured in 11 points along the three distinctive lines and in 4 points in the base material area. The lines were placed according to following manner: on the base material surface (*h2*), one millimetre above (*h1*), one millimetre below (*h3*), and two millimetres below (*h4*). The measurement points on *h1*, *h2*, and *h3* were spanned horizontally by 0.3 mm, while on *h4*, they were spanned by 1 mm (Figure 7). The reason behind taking a multitude of measurements was that the studied filler material has non-uniform consistency due to a high amount of macrocrystalline tungsten carbides embedded in the nickel matrix (Figure 8). The *h4* measurement was performed in order to estimate the influence of hardfacing heat on base material hardness and to find potential brittle fracture locations occurring because of emerging martensitic structures.

### 2.6. Carbide Segregation

The tungsten carbides present in the cored wire may exhibit different distribution within the hardfaced coating, thus it was decided to find a qualitative factor of their segregation. The lines for hardness measurement were used as boundaries of a “box” in enclosed within the *h1* and *h2* lines (refer to Figure 7). Afterwards, the authors calculated the number of macrocrystalline carbides (in this case: bigger than 100 µm in diameter of inscribed circle) occurring in the “box” for every sample, averaged and rounded up the mean values to the closest integer (Figure 9).

### 2.7. Statistical Background

The results were subjected to a two-way ANOVA test with repetitions, executed in STATISTICA 13.3 software (TIBCO Software Inc., Palo Alto, CA, USA), which helped to establish whether the volumetric flow rate (or linear energy coefficient) and the voltage level have influence on the bead geometry and hardness for each of the hardfaced materials. Additionally, this statistical tool helps to discover, whether there are interactions between the input parameters. The confidence level in this case was set to α = 0.05.

## 3. Results and Discussion

### 3.1. Chemical Composition Analysis

The results of the chemical analysis of the arc-molten wire samples are presented in Table 4. 

This method does not detect lightweight elements, such as carbon or boron, yet this analysis confirmed the high content of tungsten and nickel.

### 3.2. Heat Input Calculation

The heat input parameter is a factor of voltage, current (thus also indirectly the wire feed) divided by welding speed. It is also recognized as the linear energy coefficient modified by the welding method efficiency factor and can be calculated using Equation (2):(2)$$Q=\eta \times \frac{\int_{0}^{t}(I\times U)\;dt}{t\times v_{b}}$$

Or, to normalize the units one can apply Equation (3):(3)$$Q=\eta \times \frac{I\times U\times 60}{v_{b}\times 1000}$$
where:*Q*—heat input [kJ/mm],*I*—current [A],*U*—voltage [V],*v_b_*—welding speed [mm/min],*η*—welding efficiency factor (in the case of GMAW/FCAW it is equal to 0.8).


The formula above (3) was used to calculate the heat input level for every run (Table 5). 

The heat input level was then compared to the measurements of hardness and geometry in order to find the potential relationship between inspected factors. As Formula (3) has the welding speed in its denominator, the higher the deposition rate is applied, the lower is the calculated heat input.

### 3.3. Weld Bead Cross-Sectional Geometry Measurements

As stated in Section 2.1, the dimensional measurements were performed on 4 cut-out faces for every steel bar. The results are aggregated in Table 6 and later presented visually both with microscopic images and conventional charts. Samples from run R8 exhibit lack of penetration (Figure 10), but for calculation purposes, the appropriate values were substituted with small values (such as 0.001 mm root depth). Averaged values can be found in the respective bottom row of every run, marked with bold font.

The microscopic images were additionally used to determine the overall weld bead quality and carbide segregation. Figure 11 presents an exemplary microscopic image of each representative run. 

The polished surface of the weld bead cross-sections revealed that even though the precursory visual inspection revealed potential welding defects only in samples R8 (1000/8, 19 V), the microscopic images show some porosities and differences in tungsten carbide distribution. 

The most desirable arrangement is to have carbides evenly spread across the whole cross-sections, which occurred in R4, R9, R2, and mostly in R1. In the case of R3, R6, R7, and R5, the metal puddle was enough hot to let the carbides sink and agglomerate on the bottom of the weld. Some cross-sectional samples of R2 run also exhibited a tendency to sediment an observable amount of the dense tungsten carbide macrocrystals at the root. This could potentially lead to premature wear of a hardfaced tool, since not only would such a coating wear rapidly in the area above the tool’s nominal dimension, but also the closely-packed carbides may exhibit additional, undesired brittleness.

The height and root depth measurement of the studied weld beads revealed that increasing the voltage increases the penetration and fusion, while it decreases the height by a small value (Figure 12), simultaneously broadening the given weld bead (Figure 13). The increase of height in the case of R8 (1000/8, 24 V) is caused by lack of observed penetration and fusion between the distinctive materials.

The geometrical measurements showed that there is a strong correlation between the weld bead width and experimental setup—the slower the hardfacing process is, the bead width is less reliant on the voltage level at a rate.

### 3.4. Hardness Evaluation

The HV05 Vickers hardness was measured according to scheme presented in Section 2.5. The results are presented in two ways—numerical and visual. The visual presentation consists of color-scaled blocks, with distinctively visible, dark-tinted carbides. The averaged hardness value for every run is presented below every pair of hardness maps (Figure 14).

The hardness of the base material was measured in every sample approx. 1 mm below the border of the Heat Affected Zone in four distinctive points and was assessed to be about 200–210 HV. Yet, on the heatmap, some samples (e.g., one of 500/4, 19 V or 1000/8, 24 V) have higher hardness near that location—this indicates deep penetration along with carbides concentrated near the root.

The measurements of the first group (500/4) have the least amount of spotted carbides, which corresponds to the visual analysis of TC segregation, while the average hardness for this group was approx. 503.23 HV05. The second group (750/6) had the highest probability of encountering a hard carbide and the highest registered average hardness at approx. 514.89 HV05. The third group (1000/8) also had high probability of spotting a carbide crystal, however, due to the incomplete fusion and root carbide segregation in R5, the average hardness was the lowest observed—488.12 HV05.

### 3.5. Carbide Distribution 

This section describes the measurements of carbide distribution among different cross-sections according to the method mentioned in Section 2.7. Although some samples may have more visible carbides, they have been sunk into the root, thus they were classified as being out of the usable scope (Table 7).

### 3.6. Statistical Background

The measurement results have been subjected to the two-way ANOVA test with repetitions. The summary of the statistical tests is available in Table 8 and in Figure 15, Figure 16 and Figure 17. The values have been rounded in order to increase readability. 

The interpretation of the two-way ANOVA is the following:−There is no significant statistical difference between the dependent factors (hardfacing parameters) and the height of the bead. In the case of volumetric feed, it can be observed that the *p*-value is close to alpha. This parameter may result in a greater scatter of data under the confidence level. It is observed for the 19 V and volumetric feed 1000/8 where the value for the bead height is the greater than the rest, but still this result is not significantly different than the rest (Figure 15).−The width is affected both by voltage and the volumetric feed and there is an interaction between those two parameters in a statistical meaning. An increase in the width of the weld bead can be observed. At the same time, a higher volumetric feed value results in a lower width of the weld bead for lower voltages, which confirms the observed relationship to the value of heat input. For the two first values of volumetric feed, there are no significant differences between the weld bead width. However, for the value of 750/6, an increase in bead width is observed with increasing voltage (Figure 16).−The root depth is strongly dependent on the input voltage level and volumetric feed level and those parameters interact between one another. The observed effect of voltage and volumetric feed on penetration, in the main, is based on a clear interaction between these parameters. For the lowest voltage, there are no statistical differences between root depth and volumetric feed. Observable differences occur for higher voltage levels. For 1000/8 volumetric feed, a clear increase in penetration depth is seen (Figure 17).

As the studied wire consists of tungsten carbides and the nickel matrix, characterized by the significantly different hardness, the hardness values for the weld bead are highly scattered. The observed differences of the measured hardness of the critical area of the bead rely rather on tungsten carbide segregation, which on the other hand, is affected by the heat input level, being a combination of range of technological parameters of the process. Due to significant scatter of hardness, the nonparametric Kruskal–Wallis test for the ranks was performed to investigate the influence of the hardfacing process parameters at this property. The corelation between the hardness, voltage level, and volumetric feed is presented in Figure 18. 

The results of the rang tests for the hardness are presented in Table 9. For the Kruskal–Wallis Rang ANOVA test, only a statistical difference between the samples could be observed for the influence of volume feed. The voltage level had no significant impact on the hardness of tested materials. The post-hoc test for multiple comparisons showed that a statistically significant difference occurred between 750/6 and 1000/8 (Table 10). 

However, by analyzing the median hardness for volumetric flow, it may be observed that the K–W test results for this parameter can be associated with a significantly higher scatter of hardness for 750/6 (Figure 19). This could be related to randomly scoring a larger tungsten carbide by the indenter, which significantly moved the average of the results to higher values. At the same time, the highest number of carbides per box in the presented quantitative description of carbide segregation is observed for the 750/6 and 21.5 V parameter (Table 7).

### 3.7. Discussion

The overall and beneficial hardness—along with the hardness of the nickel matrix and the high-scoring WC spots—is dependent on the heat input. The samples which were manufactured with heat input exceeding 0.39 kJ/mm—either the ones applied slowly and the ones with high voltage and high current (resulting from high wire feed)—had the possibility to solidify for longer time, hence the large WC crystals had the opportunity to sink to the root. The best set of parameters for this wire is the one that keeps heat input at about 0.33–0.38 kJ/mm. Below those values, the material emits much spatter and dilutes with the substrate poorly. 

The studied material is relatively easy to apply on the C45 steel at the room temperature, even though the carbon equivalent of this steel is 0.52–0.82—depending on the manufacturer and C_e_ calculation standard. The hardfaced coating applied in the imperfect manner may still be usable, nonetheless the behavior and wear mechanism may differ from expectations. Compared to the tungsten carbides, the nickel matrix itself has observably lower abrasion resistance. This may lead to premature and unexpected wear of the coated tool’s working surface. After superficial abrasion, the concentrated tungsten carbide crystals may exhibit desired wear resistance, yet due to their brittleness, the aforementioned resistance may not last long. 

When it comes to the dimensional stability of the weld bead at different heat input fabrication levels, the experiment showed that the faster the deposition rate is (quotient of wire feed and welding speed), the less prone to voltage adjustment is the width. Width and penetration were the two weld bead properties, which were mostly sensitive to the change of input parameters The height of the bead seems to be highly resistant to change in parameters at the given range.

## 4. Conclusions

This paper describes the behavior of Φ 1.6 mm Ni-WC-based cored hardfacing wire applied on C45 steel with high deposition rates. Deposition rates resulting in high quality overlays were higher than in the study of Chai et al. [30]. Hence, higher process efficiency preserving the coating quality is possible.

The main conclusions of this study are listed below:The separate hardness values of both filler wire components (nickel and tungsten carbide crystals) were not affected by the heat input directly.However, when using this 1.6 mm cored wire, the user should refrain from exceeding 0.4 kJ/mm of heat input, since the molten mixture in the welding puddle cools down at such a slow rate, that the heavy tungsten carbide crystals (with density at approx. 14,500 kg/m^3^) are able to sediment at the root of nickel matrix (density at approx. 8900 kg/m^3^), rendering the hardfaced layer less reliable.For most applications, the heat input between 0.34–0.39 kJ/mm may produce the desired results for the given filler material. The dilution at that level ranges between 10% and 20% and as it provides strong connection between the substrate and the overlay, its low depth prevents the WC crystals from root aggregation.This wire is dimensionally stable, compared to other hardfacing materials studied by the authors before [28]. Its usual behavior is to form a wide and flat bead with a 4–5% height-to-width ratio.

## Figures and Tables

**Figure 1 materials-16-03960-f001:**
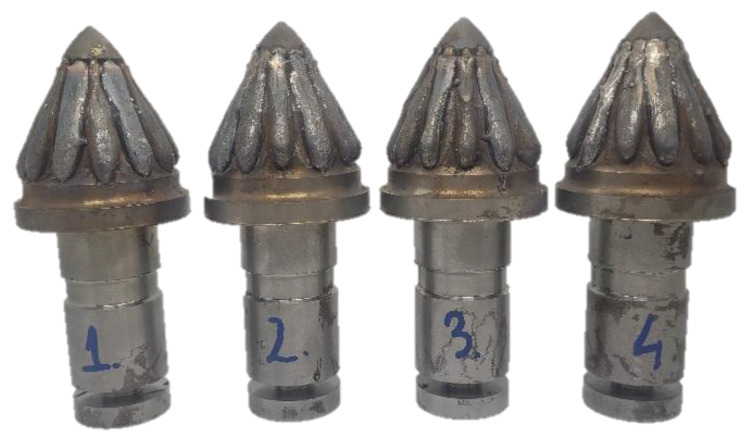
Hard coating applied with robotized GMAW process on the working surface of a mining conical pick.

**Figure 2 materials-16-03960-f002:**
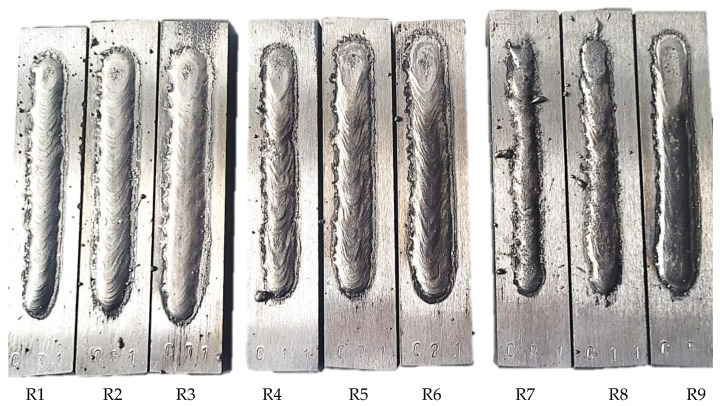
Samples R1–R9.

**Figure 3 materials-16-03960-f003:**
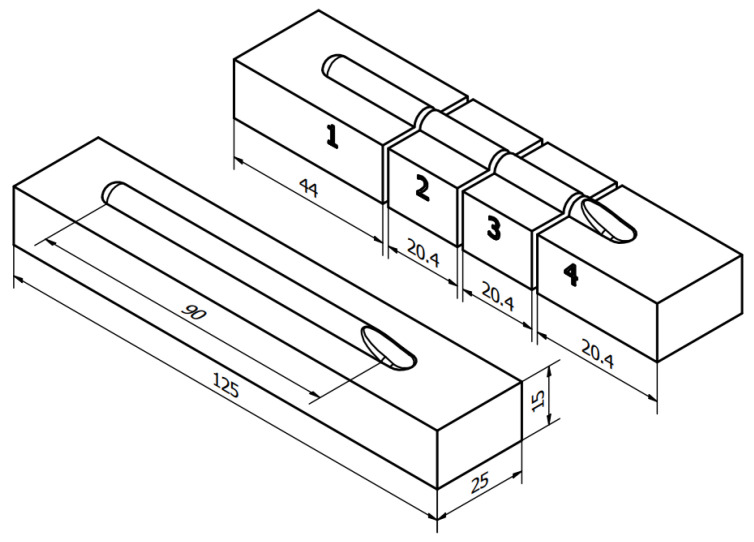
Sample preparation scheme. Cut-outs 2 and 3 were polished and taken into account in the measurements.

**Figure 4 materials-16-03960-f004:**
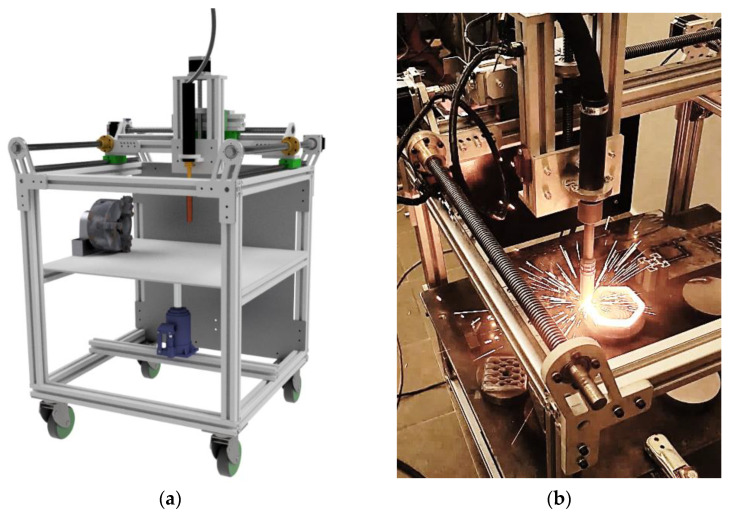
View of the CNC Hardfacing Machine, codenamed NaPawlik v2.0. (**a**) Depicts the CAD model, (**b**) shows the device in action.

**Figure 5 materials-16-03960-f005:**
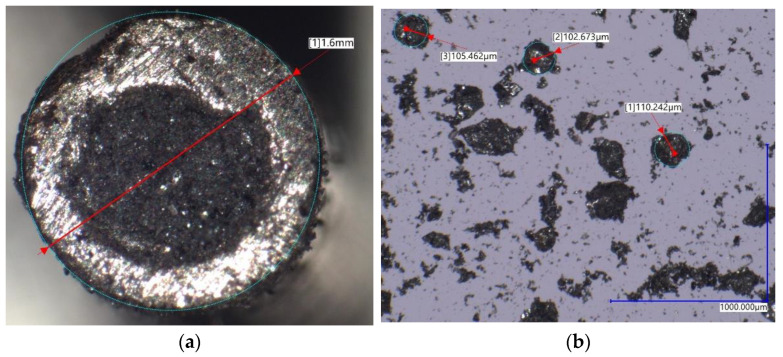
Microscopic view of the filler material cross section: (**a**) depicting the MTCs sheathed within nickel tube, (**b**) view of the exemplary tungsten carbide crystals size.

**Figure 6 materials-16-03960-f006:**
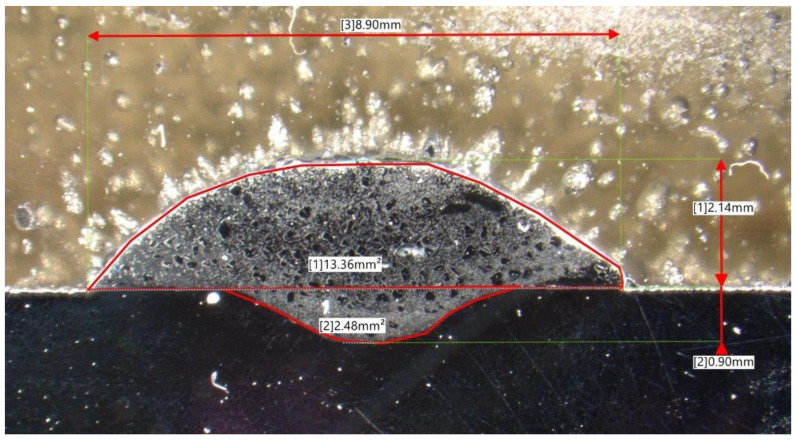
An exemplary microscopic view on the cross-sectional geometry of the studied weld bead.

**Figure 7 materials-16-03960-f007:**
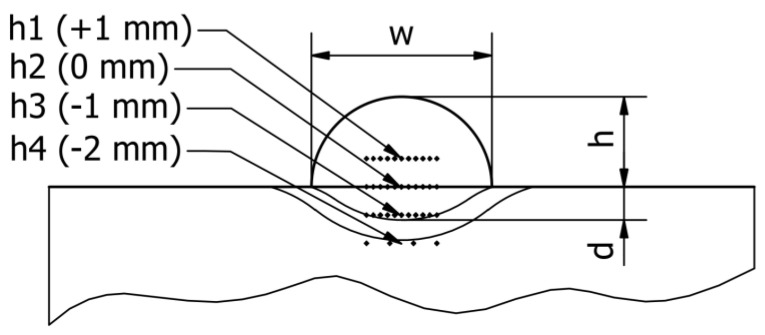
Hardness measurement scheme.

**Figure 8 materials-16-03960-f008:**
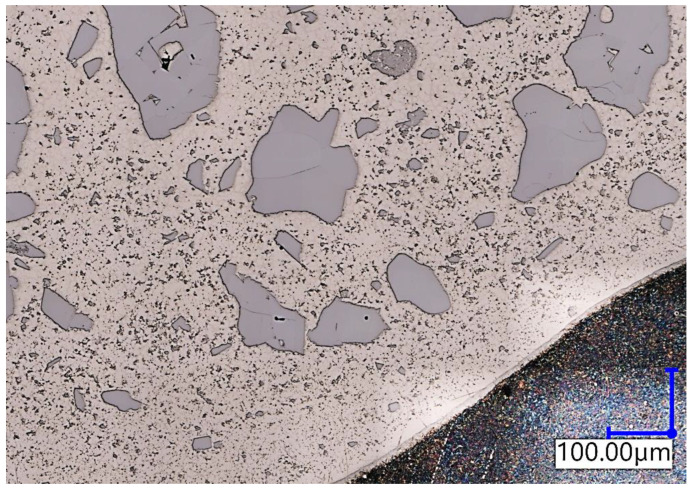
Non-uniform distribution of carbides in the investigated sample. Light grey areas—carbides, beige area—nickel matrix, dark color (bottom right)—steel substrate material.

**Figure 9 materials-16-03960-f009:**
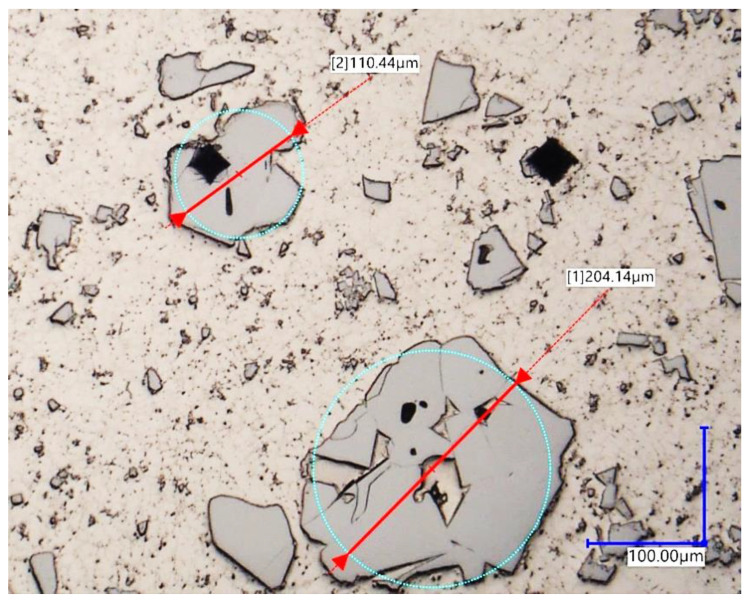
Microscopic image of carbides, classified in this study as “large”.

**Figure 10 materials-16-03960-f010:**
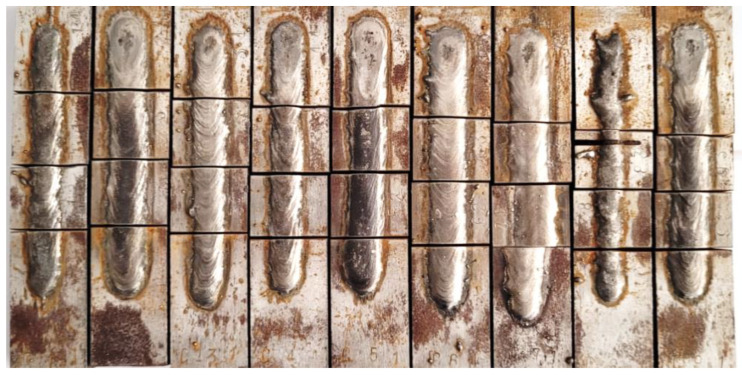
Image of the cut samples. Initial visual inspection shows that on the superficial level, only sample R8 (second from right side) may exhibit some welding defects due to heat input being too low.

**Figure 11 materials-16-03960-f011:**
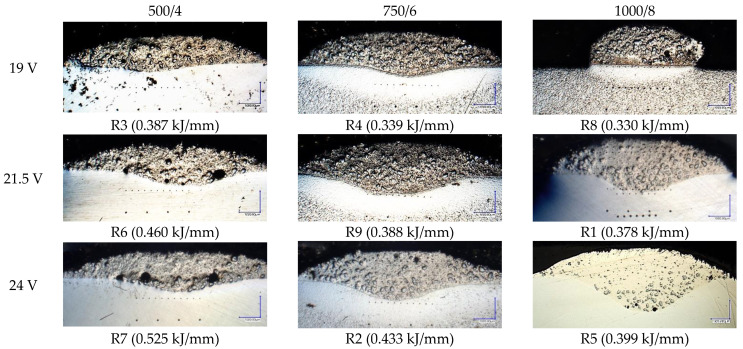
A set of representative microscopic pictures of inspected coatings cross-sections captured with 30× magnification. The brackets contain measured heat input level. The blue XY scale bar represents 1000 µm.

**Figure 12 materials-16-03960-f012:**
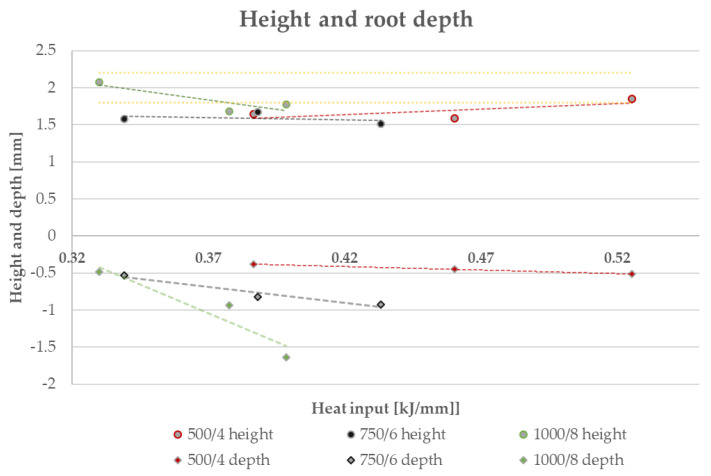
Graph representing the dependence of weld bead height and penetration (marked as “depth”) on the heat input level. The yellow dashed lines represent the assumed weld bead height.

**Figure 13 materials-16-03960-f013:**
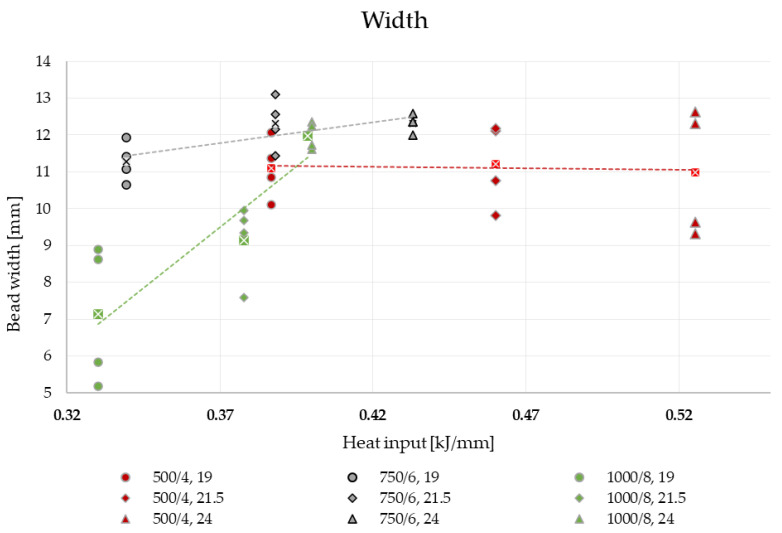
Graph representing the dependence of weld bead width on the heat input and voltage level.

**Figure 14 materials-16-03960-f014:**
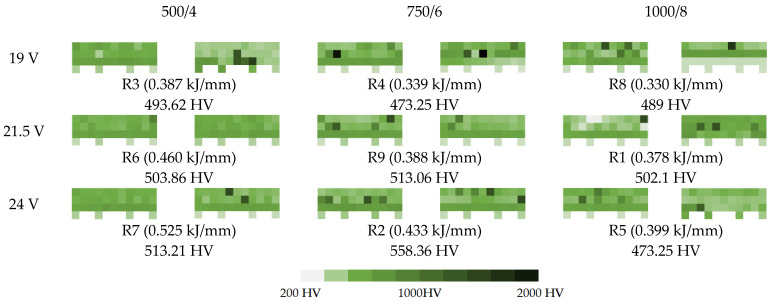
A set of hardness maps, where lighter color means lower hardness and darker color means higher hardness. The dark-green squares are the spots where the hardness tester hit a MTC (Macrocrystallite Tungsten Carbide).

**Figure 15 materials-16-03960-f015:**
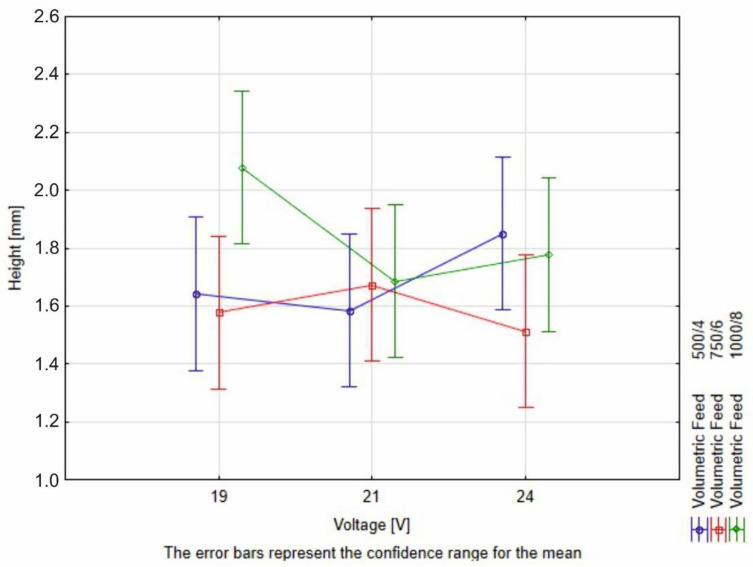
Illustration of resistivity of the bead height to the volumetric deposition feed and voltage adjustments.

**Figure 16 materials-16-03960-f016:**
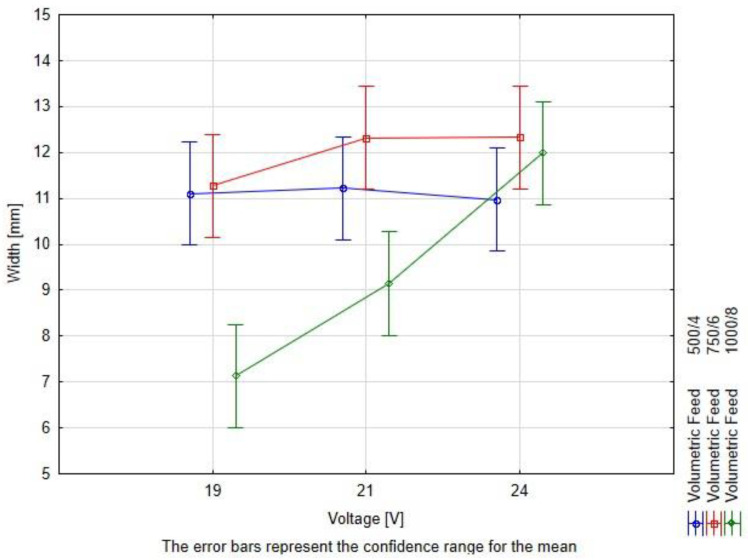
Illustration of dependence of the hardfaced overlay width on the voltage and volumetric feed. As the volumetric flow decreases, the effect of the voltage level is weaker and vice versa.

**Figure 17 materials-16-03960-f017:**
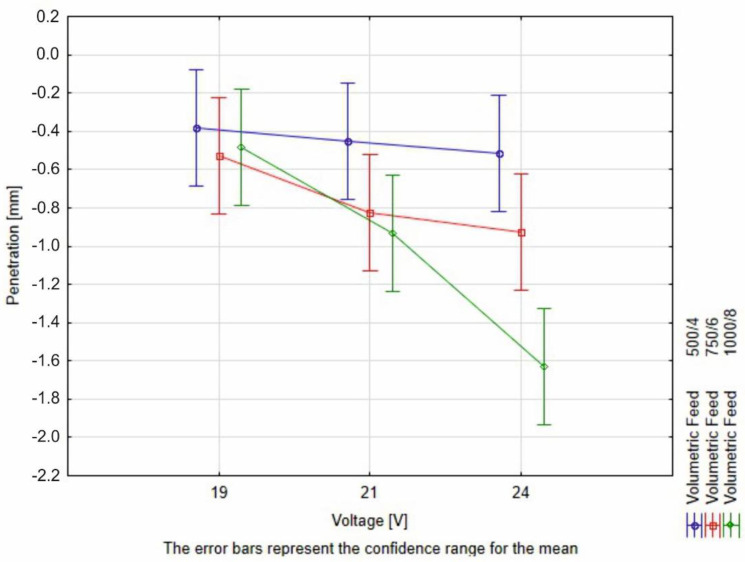
Illustration of dependency of root depth on the volumetric deposition feed and voltage levels. This graph shows, that along with the increase of the welding speed, the role of the voltage intensifies.

**Figure 18 materials-16-03960-f018:**
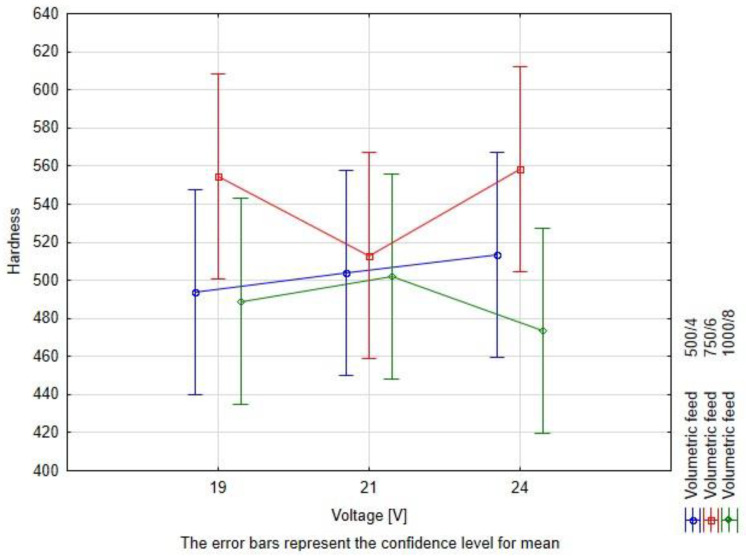
Illustration of lack of any observable dependency on the average hardness of both volumetric feed and voltage level.

**Figure 19 materials-16-03960-f019:**
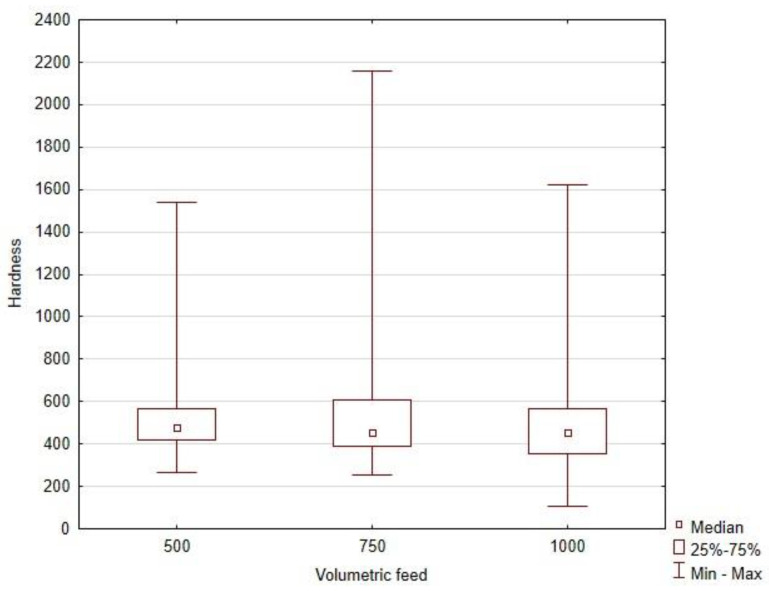
Box plots for medians, quartiles, and scatter of the hardness depending on a volumetric feed value.

**Table 1 materials-16-03960-t001:** Chemical composition of the C45 steel, as provided by the supplier.

Chemical Composition	%
Carbon	0.42–0.5
Manganese	0.5–0.8
Silicon	0.1–0.4
Phosphorus	max. 0.04
Sulfur	max. 0.04
Chromium	max. 0.3
Nickel	max. 0.3
Molybdenum	max. 0.1
Copper	max. 0.3
Fe	bal.

**Table 2 materials-16-03960-t002:** Chemical composition of the hardfacing wire used in this experiment. Those values are as declared by the manufacturer [36].

Chemical Component	Declared Value [%]
Carbon	0.3
Silicon	0.5
Boron	1.00
WC	45.0
Nickel	Bal (53.2)

**Table 3 materials-16-03960-t003:** Values for wire feed and welding head velocity selected for both kinds of materials.

Run	Wire Feed [m/min]	Head Velocity [mm/min]	Volumetric Flow [-]	Voltage [V]	Voltage Level [-]
R1	8	1000	3	21.5	2
R2	6	750	2	24	3
R3	4	500	1	19	1
R4	6	750	2	19	1
R5	8	1000	3	24	3
R6	4	500	1	21.5	2
R7	4	500	1	24	3
R8	8	1000	3	19	1
R9	6	750	2	21.5	2

**Table 4 materials-16-03960-t004:** The chemical composition of the investigated cored wires (optical emission spectroscopy).

Sample	Si	Ni	W	Fe
Sample 1	-	-	100%	-
Sample 2	1.66%	62.55%	23.69%	12.1%
Sample 3	-	61.23%	7.34%	31.43%
Sample 4	0.44%	41.2%	36.22%	22.14%
Sample 5	0.88%	76.41%	15.88%	6.82%
Mean value	0.59%	48.28%	36.62%	14.5%

**Table 5 materials-16-03960-t005:** Heat input calculated for every fabricated stringer bead.

Run Number	Welding Speed *v_b_* [mm/min]	Voltage [V]	Measured Current [A]	Calculated Heat Input [kJ/mm]
1	1000	21.5	366	0.378
2	750	24	282	0.433
3	500	19	212	0.387
4	750	19	279	0.339
5	1000	24	346	0.399
6	500	21.5	223	0.460
7	500	24	228	0.525
8	1000	19	362	0.330
9	750	21.5	282	0.388

**Table 6 materials-16-03960-t006:** Results of geometric measurements. Please note, that the depth *d* = 0.001 mm and penetration area *A_p_* = 0.00001 was written only for calculation purposes, and in reality, was close to zero.

Run #	Sample	Heat Input [kJ/mm]	Width *w* [mm]	Height *h* [mm]	Depth *d* [mm]	Bead Area *A_n_* [mm^2^]	Penetration Area *A_p_* [mm^2^]	Dilution [%]
1	1	0.378	9.95	2.15	0.31	14.79	0.54	3.52%
	2		7.59	0.97	1.5	4.42	6.46	59.38%
	3		9.35	2.05	0.9	13.75	3.38	19.73%
	4		9.69	1.57	1.03	12.71	4.65	26.79%
	**Average**		**9.145**	**1.685**	**0.935**	**11.4175**	**3.758**	**27.35%**
2	1	0.433	12.59	1.68	0.85	13.95	2.76	16.52%
	2		12.36	1.39	0.94	11.71	3.14	21.14%
	3		12.36	1.45	0.97	12.04	3.53	22.67%
	4		11.99	1.53	0.95	12.03	3.56	22.84%
	**Average**		**12.325**	**1.5125**	**0.9275**	**12.4325**	**3.248**	**20.79%**
3	1	0.387	11.38	1.58	0.49	12.32	1.8	12.75%
	2		12.07	1.56	0.69	12.72	2.44	16.09%
	3		10.85	1.68	0.34	12.63	0.91	6.72%
	4		10.11	1.75	0.001	12.3	0.00001	0.00%
	**Average**		**11.1025**	**1.6425**	**0.38025**	**12.4925**	**1.286**	**8.89%**
4	1	0.339	10.66	1.81	0.52	13.76	1.46	9.59%
	2		11.07	1.65	0.57	12.22	1.42	10.41%
	3		11.41	1.44	0.47	10.93	1.5	12.07%
	4		11.94	1.41	0.56	11.15	2.18	16.35%
	**Average**		**11.27**	**1.5775**	**0.53**	**12.015**	**1.64**	**12.11%**
5	1	0.399	12.37	1.73	1.78	13.87	6.99	33.51%
	2		12.25	1.91	1.28	15.43	5.29	25.53%
	3		11.61	1.69	1.74	13.12	7.63	36.77%
	4		11.72	1.78	1.73	13.45	7.03	34.33%
	**Average**		**11.9875**	**1.7775**	**1.6325**	**13.9675**	**6.735**	**32.53%**
6	1	0.460	10.76	1.82	0.2	13.28	0.45	3.28%
	2		9.82	1.67	0.41	11.14	1.19	9.65%
	3		12.12	1.48	0.66	12.24	2.42	16.51%
	4		12.19	1.37	0.53	11.21	2.2	16.41%
	**Average**		**11.2225**	**1.585**	**0.45**	**11.9675**	**1.565**	**11.46%**
7	1	0.525	12.64	1.49	0.62	12.98	2.15	14.21%
	2		12.32	1.61	0.23	13.79	0.82	5.61%
	3		9.63	1.84	0.47	12.89	1.08	7.73%
	4		9.32	2.46	0.73	14.2	2.67	15.83%
	**Average**		**10.9775**	**1.85**	**0.5125**	**13.465**	**1.68**	**10.85%**
8	1	0.330	8.9	2.14	0.9	13.36	2.48	15.66%
	2		5.83	2.02	0.001	9.24	0.00001	0.00%
	3		5.18	2.1	0.001	9.63	0.00001	0.00%
	4		8.63	2.05	1.04	12.2	3.81	23.80%
	**Average**		**7.135**	**2.0775**	**0.4855**	**11.1075**	**1.572505**	**9.86%**
9	1	0.388	12.16	1.72	0.85	14.19	4.02	22.08%
	2		12.56	1.75	0.9	13.8	3.33	19.44%
	3		11.43	1.69	0.75	13.12	3.02	18.71%
	4		13.11	1.53	0.79	13.62	3.73	21.50%
	**Average**		**12.315**	**1.6725**	**0.8225**	**13.6825**	**3.525**	**20.43%**

**Table 7 materials-16-03960-t007:** Stitched microscopic images presenting carbide distribution of samples R1–R9. Blue XY bars represent 1000 µm.

	500/4	750/6	1000/8
19 V	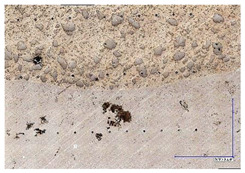	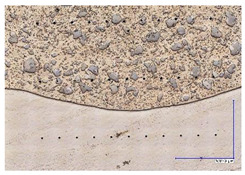	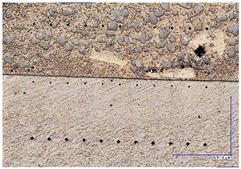
	R3 (0.387 kJ/mm) Average amount of MTC: 38	R4 (0.339 kJ/mm) Average amount of MTC: 54	R8 (0.330 kJ/mm) Average amount of MTC: 47
21.5 V	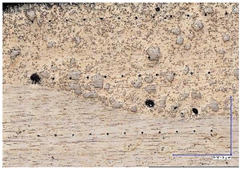	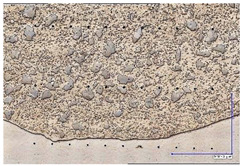	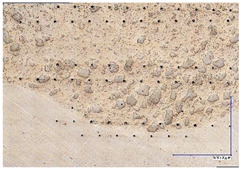
	R6 (0.460 kJ/mm) Average amount of MTC: 21	R9 (0.388 kJ/mm) Average amount of MTC: 49	R1 (0.378 kJ/mm) Average amount of MTC: 26
24 V	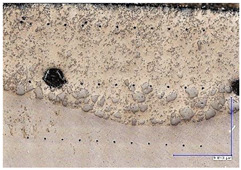	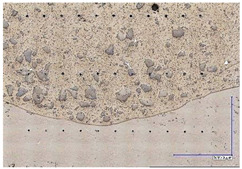	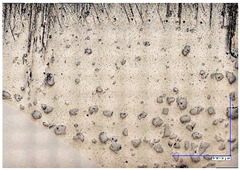
	R7 (0.525 kJ/mm) Average amount of MTC: 17	R2 (0.433 kJ/mm) Average amount of MTC: 24	R5 (0.399 kJ/mm) Average amount of MTC: 16

**Table 8 materials-16-03960-t008:** Aggregated results from two-way ANOVA test with repeated observations. “F” symbol stands for calculated F-statistic parameter, while *p*-value stands for probability factor.

		SS	df	MS	F	*p*-Value	F Crit
Height	Voltage	0.08	2	0.04	0.63	0.5379	3.35
	Volumetric feed	0.41	2	0.20	3.07	0.0630	3.35
	Interaction	0.46	4	0.11	1.73	0.1728	2.73
	Within	1.80	27	0.07			
Width	Voltage	22.36	2	11.18	9.32	0.0008	3.35
	Volumetric feed	40.25	2	20.12	16.78	0.0000	3.35
	Interaction	28.25	4	7.06	5.89	0.0015	2.73
	Within	32.39	27	1.20			
Penetration	Voltage	1.87	2	0.94	10.67	0.0004	3.35
	Volumetric feed	1.96	2	0.98	11.13	0.0003	3.35
	Interaction	1.17	4	0.29	3.33	0.0242	2.73
	Within	2.37	27	0.09			

**Table 9 materials-16-03960-t009:** Kruskal–Wallis rank ANOVA test results for the hardness of investigated materials.

Kruskal–Wallis Rang ANOVA: Hardness Independent Variable: Voltage [V] Kruskal–Wallis Test: H _(df=2, N=594)_ = 2.1099; *p*-Value = 0.3482			
Variable	N-valid	Sum of ranks	Average rank
19.0 V	198	56,115	283.4
21.5 V	198	60,860	307.4
24.0 V	198	59,740	301.7
Kruskal–Wallis Rang ANOVA: Hardness Independent variable: Volumetric feed Kruskal–Wallis test: H _(df=2, N=594)_ = 7.0454; *p*-Value = 0.0295			
Variable	N-valid	Sum of ranks	Average rank
500/4	198	60,043.5	303.3
750/6	198	62,759.5	316.9
1000/8	198	53,912.0	272.3

**Table 10 materials-16-03960-t010:** The *p*-value for multiple comparisons (two-sided). Kruskal–Wallis test.

Hardness	500/4 R: 303.3	750/6 R: 316.9	1000/8 R: 272.3
500/4	-	1.0000	0.2178
750/6	1.0000	-	0.0287
1000/8	0.2178	0.0287	-

## Data Availability

The data presented in this study are available on request from the corresponding author.

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
