# Peer review of "On the Influence of Heat Input on Ni-WC GMAW Hardfaced Coating Properties"

_materials, 2023, doi:10.3390/ma16113960_

Round 1
Reviewer 1 Report
Line 134: To be used
Line 135: parts are
Line 150: delete basing
Line 163: “This formula above” replace with Equation 1
Line 170-172: Sentence should be revised
Line 175: Delete the at the beginning of the sentence
Line 185: Revise “In the aspect of…”
Line 195-196: Revise
Line 214: Revise “In enclosed within”
Line 217: Revise and specify what you mean by “Rouded up to the values”
Line 224: Delete does or does not
Line 230: Delete below
Line 233: Delete allow to
Line 238: Replace the formula below with Equation 2.
Line 239: Replace following version with Equation 3
Line 246: Replace formula above with Equation 3
Line 256: “As stated in Chapter 2.6” refer to the correct section
Line 267: delete “the” in the figure 10 and gallery
Line 272: Revise “Drawn attention only”
Line 273: shown
Line 277: Revise “enough hot”
Line 279: TC is not the abbreviation for tungsten carbide
Line 280-281: Revise
Line 292: Shows instead of shown
Line 294: Revise
Line 322: Revise “thus being”
Line 369: Correlation spelling
Line 374: Revise “at Table 9”
Line 378: Why is table abbreviated?
Line 394: Revise “shown”
Line 401: At room temperature and not in room temperature
Line 407: TC
General comments
· Note: Equation numbering should be corrected, there are 2 equation 1’s
· Image R5 in Figure 11 has visible scratches, prepare the sample making sure scratches are not prominent
· Why is tungsten carbide abbreviated TC and not WC?
· Fit Table 1 in one page,
· Results and discussion should be supported by literature review
· Draw conclusions from the study, do not confuse the discussion with conclusions
· Include volume, page numbers, and journal name for reference 9
· Reference 20 and 21, Journal names missing, title in reference 21 is in capital letters, use lower cases
- The use of “the Authors” throughout the article should be reconsidered. The work was done by the authors, and therefore it is not necessary to constantly and repeatedly refer to Authors.
-Authors should pay attention to the use of tense
-Pay attention to numbering of equations, and delete parameters given in lines 158-160.
-The manuscript should be thoroughly edited
- Correlating results with previous studies will add value to the work. References were not adequately used to support the study
-Refence style used in the reference list should be consistent throughout.
-Abstract should be revised to clearly highlight the aim, objectives, procedure, results and conclusions.
Author Response
Dear Reviewer,
We would like to thank you for the careful and thorough review. We addressed all of the language issues that you have listed. We would like to reply to the general comments below:
Remark 1
Note: Equation numbering should be corrected, there are 2 equation 1’s
Answer: Thank you, the equation numbers are now correct.
Remark 2
Image R5 in Figure 11 has visible scratches, prepare the sample making sure scratches are not prominent
Answer: We have re-captured the microscopic image of R5 after additional round of polishing.
Remark 3
Why is tungsten carbide abbreviated TC and not WC?
Answer: You are right – tungsten carbides are formally called WC, not TC. One of the authors was convinced otherwise due to frequent use of “TC” abbreviation in market offers.
Remark 4
Fit Table 1 in one page,
Answer: We promise that we will apply special care to the arrangement of all the tables, image descriptions etc. If you decide to approve our manuscript for publishing, they will be fit appropriately on the Editor phase. For now, we leave the manuscript in change-tracking mode.
Remark 5
Results and discussion should be supported by literature review
Answer: We added reference to similar study, where the authors used similar materials, but their coatings were manufactured with lower kg/h efficiency.
Remark 6
Draw conclusions from the study, do not confuse the discussion with conclusions
Answer: We believe it is now correct.
Remark 7
Include volume, page numbers, and journal name for reference 9
Answer: It is fixed.
Remark 8
Reference 20 and 21, Journal names missing, title in reference 21 is in capital letters, use lower cases
Answer: It is now fixed.
We also addressed the remark about too frequent use of “the Authors” – it is now fixed. We also revised the abstract and edited the paper where it was needed. We hope you will find our manuscript suitable for publishing. Best regards,
Authors.
Reviewer 2 Report
Journal: Materials (ISSN 1996-1944)
Manuscript ID: materials-2392048
Review Report 1#
The authors presented an article on “On the influence of heat input on Ni-TC GMAW hardfaced coating properties”. The subject of the article falls within the scope of the journal "Materials". However, the article will be ready for publication after a major revision. Comments are listed below. (The similarity rate is 19%.)
1. In the introduction, the properties and usage areas of C45 steel used in the experiments were not mentioned. It should be added in detail.
2. In the last paragraph of the introduction, the difference between the study and previous studies in the literature should be clearly stated. What is the novelty in this study?
3. The parameters applied in the experiments were chosen according to which standards.
4. No reference is made to Figure 1 and Figure 3 in the text. It should be attached.
5. On page 11, line 267, "figure10" should be "figure 11".
6. Why was the Brinell method not chosen for hardness measurement? In the Brinell method, on the other hand, better results can be obtained due to the contact with a larger surface.
7. Why was protective atmosphere gas used during welding? Why was Argon gas chosen? There are other shielding gases as well.
8. What do the "F" and "p" columns in Table 8 mean? It should be explained.
9. The "contribution rates" column can be added to Table 8.
10. The Results section is devoid of discussion. Similar studies in the literature should be discussed in more detail by giving examples.
11. On page 19, line 393, it should be "Conclussions" instead of "Discussions".
12. The article contains numerous typographic and language errors. It should be corrected.
13. The article should be rearranged by taking into account the journal writing rules and citation rules.
*** Authors must consider them properly before submitting the revised manuscript. A point-by-point reply is required when the revised files are submitted.

Author Response
Dear Rewiever,
We would like to thank you for the careful and thorough review. Thank you for your remarks. Please allow us to address your remarks:
Remark 1
In the introduction, the properties and usage areas of C45 steel used in the experiments were not mentioned. It should be added in detail.
Answer: We included the applications of C45 steel to the introduction section.
Remark 2
In the last paragraph of the introduction, the difference between the study and previous studies in the literature should be clearly stated. What is the novelty in this study?
Answer: The main novelty is the set of parameters that result in better properties of hardfaced overlay manufactured at higher deposition rate, which might be useful in robotized hardfacing process and also could increase the kg/h efficiency.
Remark 3
The parameters applied in the experiments were chosen according to which standards.
Answer: Parameters were set according to the range recommended by the manufacturer. It appeared though, that the highest value of voltage may produce unreliable overlays.
Remark 4
No reference is made to Figure 1 and Figure 3 in the text. It should be attached.
Answer: It is now fixed.
Remark 5
On page 11, line 267, "figure10" should be "figure 11".
Answer: It is now fixed, thanks for the remark.
Remark 6
Why was the Brinell method not chosen for hardness measurement? In the Brinell method, on the other hand, better results can be obtained due to the contact with a larger surface.
Answer: The outcome of Brinell method may return only averaged values of hardness. We intended to measure the hardness of two wire constituents separately. We added proper explanation to the manuscript.
Remark 7
Why was protective atmosphere gas used during welding? Why was Argon gas chosen? There are other shielding gases as well.
Answer: We used argon with no admixtures, since it is most commonly used gas in inert hardfacing.
Remark 8
What do the "F" and "p" columns in Table 8 mean? It should be explained.
Answer: Those are symbols for calculated Fisher statistic factor and probability value. It is now included in the Table description.
Remark 9
The "contribution rates" column can be added to Table 8.
Answer: We are sorry, but we would be happy if you elaborate on what exactly do you mean by “contribution rates”?
Remark 10
The Results section is devoid of discussion. Similar studies in the literature should be discussed in more detail by giving examples.
Answer: It is now fixed – both in introductory section and in results section.
Remark 11
On page 19, line 393, it should be "Conclussions" instead of "Discussions".
Answer: It is fixed now, thanks for the remark.
Remark 12
The article contains numerous typographic and language errors. It should be corrected.
Answer: We believe it is now fixed.
Remark 13
The article should be rearranged by taking into account the journal writing rules and citation rules.
Answer: We believe it is now fixed, we also corrected some references (there were missing journal names, pages etc).
We hope all your remarks are properly addressed. We enclose best regards,
Authors.
Reviewer 3 Report
This manuscript investigates the influence of heat input on Ni-TC GMAW hardfaced coating properties. The materials substrate was a C45 steel. The authors investigated a welding speed range from 500 to 1000 mm/min, a voltage range from 19 to 24 V, a current range from 212 to 366 A, a wire feed range from 4 to 8 m/min. The final calculated heat input was in the range 0.330-0.525 kJ/mm. In my opinion this work can be accepted after major revisions, which are listed in the following.
The abstract should not have the period “The heat input is a value resulting from … lower limit for the studied wire.”, because this is part of the materials and methods section. The final part of the introduction should report the aim of the paper preferably using a list to clearly state the goal of the work. The manuscript does not have the conclusion section.
A limited variation of the weld bead height, penetration and width can be observed (see Figure 12 and 13). A clear trend is visible only when the welding speed is 1000 mm/min and the feed rate is 8 m/min. On the other hand, the heat input and the weld bead microhardness show the larger variation when the welding speed is 500 mm/min and the feed rate is 4 m/min (see Figure 14). The discussion section should focus of which set of process parameter should be preferred and a wear test should be carried out to achieve this target.
Author Response
Dear Reviewer,
We would like to thank you for the careful and thorough review. We would like to elaborate on the final remark:
Remark 1
The abstract should not have the period “The heat input is a value resulting from … lower limit for the studied wire.”, because this is part of the materials and methods section. The final part of the introduction should report the aim of the paper preferably using a list to clearly state the goal of the work. The manuscript does not have the conclusion section.
Answer: The abstract section is now revised according to your suggestions. We also changed “discussion” section to “conclusions” section, and followed your advice to list the most important findings.
Remark 2
A limited variation of the weld bead height, penetration and width can be observed (see Figure 12 and 13). A clear trend is visible only when the welding speed is 1000 mm/min and the feed rate is 8 m/min. On the other hand, the heat input and the weld bead microhardness show the larger variation when the welding speed is 500 mm/min and the feed rate is 4 m/min (see Figure 14).
Answer: That is our actual finding – the overall hardness (along with hardness of nickel matrix and the high-scoring WC spots) is dependant on the heat input, which is a factor of voltage, current (thus also indirectly the wire feed) divided by welding speed. The samples which were manufactured with heat input exceeding 0.49 kJ/mm –either the ones applied slowly and the ones with high voltage and high current (resulting from high wire feed) – had possibility to solidify for longer time, hence the large WC crystals had sunk to the root. The best set of parameters for this wire is the one, that keeps heat input at about 0.33-0.38 kJ/mm. Below those values the material emits much spatter and dilutes with the substrate poorly.
We have included the information above to the Results and Discussion section. Our response to the second part of final remark:
Remark 3
“The discussion section should focus of which set of process parameter should be preferred and a wear test should be carried out to achieve this target.”
Answer: When is comes to the wear test, we thought about that and we plan to carry the tests according to ASTM G65 in our future research, as a separate manuscript with comparison between resistance to abrasive wear of several hardfacing materials, that we studied. The most probable outcome for this wire is expected to be as follows:
- Samples with most of carbides “sunken” at the bead root will exhibit slightly smaller wear rate than the nickel metal itself. It will be probably better than pure nickel due to presence of smaller “floating” WC crystals.
- Samples with carbides that are distributed evenly may exhibit very high wear resistance, due to WC Mohs score being close to diamond (9-9.5). Wear of the hardfaced layer will be mostly impacted by loss of the matrix holding the crystals.
However, we would like to compare it to other hardfacing materials.
We hope that you find our corrected version of manuscript more suitable for publishing.
Best regards,
Authors.
Round 2
Reviewer 2 Report
Journal: Materials (ISSN 1996-1944)
Manuscript ID: materials-2392048
Review Report 2#
The authors made the desired corrections. In my opinion, this article can be accepted for publication in the "Materials" journal in its final form.

Author Response
Thank you.
Reviewer 3 Report
The authors have amended the required changes. The paper can be accepted in the present form.
Author Response
Thank you.